environmental chemistry

phosphate adsorption, La$_2$O$_3$, CMK-3,
solid-state grinding, confinement effects

**Author for correspondence:**
Tao Liu
e-mail: liut@njit.edu.cn

This article has been edited by the Royal Society of Chemistry, including the commissioning, peer review process and editorial aspects up to the point of acceptance.

# Confined La$_2$O$_3$ particles in mesoporous carbon material for enhanced phosphate adsorption

Xiaoqiu Ju[1], He Cui[1], Tao Liu[1,2], Yabing Sun[1], Shourong Zheng[1] and Xiaolei Qu[1]

[1]State Key Laboratory of Pollution Control and Resource Reuse, School of the Environment, Nanjing University, Nanjing 210046, People's Republic of China
[2]School of Environmental Engineering, Nanjing Institute of Technology, Nanjing 211167, People's Republic of China

(iD) TL, 0000-0002-0959-8200; SZ, 0000-0001-8660-4910;
XQ, 0000-0002-9157-4274

Novel phosphate adsorbents with confined La$_2$O$_3$ inside mesoporous carbon were fabricated by the solid-state grinding method using pristine mesoporous carbon material CMK-3 (PCMK-3) and oxidized CMK-3 (OCMK-3) as the matrixes (denoted as La$_2$O$_3$@PCMK-3 and La$_2$O$_3$@OCMK-3). Compared with pure La$_2$O$_3$, La$_2$O$_3$@PCMK-3 and La$_2$O$_3$@OCMK-3 exhibited higher normalized phosphate adsorption capacity, indicative of efficient loading of La$_2$O$_3$ inside the mesopores of the carbon materials. Furthermore, La$_2$O$_3$ loading led to substantially enhanced phosphate adsorption. The adsorption capacities of La$_2$O$_3$@OCMK-3 samples were higher than those of La$_2$O$_3$@PCMK-3 samples, possibly owing to the oxygen-containing groups forming in OCMK-3 during HNO$_3$ oxidation, which enhanced the dispersion of La$_2$O$_3$ in the mesopores of OCMK-3. The adsorption capacities of La$_2$O$_3$@PCMK-3 and La$_2$O$_3$@OCMK-3 increased with the La$_2$O$_3$ loading amount. Phosphate adsorption onto La$_2$O$_3$(14.7)@PCMK-3 followed the pseudo-second-order kinetics with respect to correlation coefficient values (larger than 0.99). As pH increased from 3.4 to 12.0, the phosphate adsorption amounts of La$_2$O$_3$(14.7)@PCMK-3 and La$_2$O$_3$(15.7)@OCMK-3 decreased from 37.64 mg g$^{-1}$ and 37.08 mg g$^{-1}$ to 21.92 mg g$^{-1}$ and 14.18 mg g$^{-1}$, respectively. Additionally, La$_2$O$_3$@PCMK-3 showed higher adsorption selectivity towards phosphate than coexisting Cl$^-$, NO$_3^-$ and SO$_4^{2-}$. The adsorbent La$_2$O$_3$(14.7)@PCMK-3 remained stable after five regeneration cycles.

# 1. Introduction

Phosphorus (P) is an essential nutrient for organism growth and a nonrenewable resource [1–3]. The natural P reserves are predicted to run out within the next few decades [4,5]. On the other hand, excess P from domestic and industrial wastewater discharged into natural water will lead to eutrophication, which has attracted worldwide attention [6–8]. It is commonly recognized that the removal and recovery of P from wastewater is an effective approach to control eutrophication and alleviate the shortage of P resources [9,10]. Therefore, it is essential to develop effective methods to remove and recover P before its discharge into natural waters. Many methods were explored to remove P from water, such as precipitation, adsorption and biological treatment [11–14]. Among them, the adsorption method has been extensively studied due to its economy, convenience and efficiency. However, most adsorbents suffered from low adsorption capacity, interference by coexisting anions and regeneration issues, limiting their application in P removal.

Metal oxide modification is believed to be a widely used method to enhance the adsorption capacity of adsorbent, such as La(III)-, Zr(IV)- and Fe(III) oxide modification [15–18]. Previous studies showed that adsorbents modified by La(III) oxide were effective for phosphate removal [19–21]. It should be emphasized that highly dispersed metal oxides/hydroxides without aggregation can provide more adsorption sites for phosphate removal [22]. Therefore, to enhance the dispersion of active species, metal oxide/hydroxides are usually loaded on various carriers with large surface areas. Due to the high surface area, large pore volume and ordered pore structure, porous materials were widely used in interface-related processes, such as catalysis [23,24], adsorption [17,22], energy storage [25–27] and so on. For instance, Zhang *et al*. [24] reported that the graphitic nitrogen-doped porous carbon-based electrocatalyst showed high electrocatalytic performance for the oxygen reduction reaction. Wei *et al*. [28] previously synthesized a flower-like material with a large surface area and rich mesoporosity, which exhibited excellent super capacitor performances. Shin *et al*. [29] reported that compared with activated alumina, Al-impregnated mesoporous material exhibited more rapid adsorption kinetics and higher phosphate adsorption capacity. Additionally, Bacelo *et al*. [30] suggested that the adsorption capacities of La-modified zeolites were 1.5–5 times higher than those of natural and synthetic zeolites without La treatment. Therefore, we believed that porous materials as carrier matrixes could effectively improve the adsorption property of active species and played a very crucial role in phosphate adsorption.

In this study, the main objective is to verify the feasibility of using mesoporous material CMK-3 to confine $La_2O_3$ particles for phosphate adsorption. The La-confined adsorbents prepared by the solid-state grinding method were well characterized by X-ray diffraction (XRD), X-ray fluorescence (XRF), transmission electron microscope (TEM), $N_2$ adsorption–desorption, Fourier transform infrared spectroscopy (FT-IR), zeta potential measurements and X-ray photoelectron spectroscopy (XPS). The adsorption behaviour for phosphate from the water was further investigated, such as adsorption kinetics, isotherms as well as the influence of pH and coexisting anions. The regeneration property of La-confined adsorbent was evaluated by consecutive adsorption–regeneration experiments for phosphate adsorption.

# 2. Experiment

## 2.1. Adsorbent preparation

### 2.1.1. Preparation of mesoporous carbons

The pristine mesoporous carbon, PCMK-3, was prepared by using SBA-15 as the hard template according to the method reported in the previous study [31]. The oxidized CMK-3 (denoted as OCMK-3) was prepared as follows: 1.0 g of PCMK-3 was mixed with 150 ml of nitric acid solution (2 M) and the mixture was oxidized at 70°C for 4 h. After reaction and cooling to room temperature, the mixture was filtrated, washed by deionized water to neutral pH and dried at 100°C for 12 h.

### 2.1.2. Preparation of La-confined adsorbents

To prepared La-confined PCMK-3, 0.02–0.06 g of $La(NO_3)_3 \cdot 6H_2O$ and 0.2 g of PCMK-3 were mixed and repeatedly ground in an agate mortar at 25°C for 30 min. The mixture was then transferred into an oven

and maintained at 30°C for 48 h. Then, the mixture was exposed to $NH_3$ atmosphere to form hydroxides in a desiccator with 200 ml of concentrated aqueous ammonia (28%) for 48 h. Finally, the mixture was washed with deionized water till pH close to 7.0 and dried at 105°C for 12 h. The resulting material was labelled as $La_2O_3(X)@PCMK-3$, where the X was the $La_2O_3$ loading amount (wt.%) measured using XRF. $La_2O_3(X)@OCMK-3$ were prepared by a similar procedure but using OCMK-3 as the matrix.

### 2.1.3. Preparation of pure $La_2O_3$

To prepare $La_2O_3$, 2.0 M ammonium hydroxide solution was added dropwise to 0.2 M $La(NO_3)_3$ solution till pH 10.5 under vigorous stirring. After stirring for 10 h at room temperature, the resulting material was filtrated and then washed by deionized water and dried at 105°C for 12 h.

## 2.2. Material characterization

The morphology of samples was observed by scanning electron microscopy (SEM, Quanta FEG 250, USA). TEM images of the samples were obtained on a TF20 electron microscope using an operating voltage of 200 kV. Small-angle XRD patterns of the samples were obtained on an X'TRA powder diffraction-meter (Thermofisher Scientific Co., USA), operating with Cu K$\alpha$ radiation (40 kV, 40 mA) over $2\theta$ range of 0.75–5. The $La_2O_3$ contents of the samples were measures on an ARL9800XP X-ray fluorescence spectrometer (Thermo Electron, Switzerland). Surface areas and pore width distributions were determined using $N_2$ adsorption–desorption isotherms at −196°C on a Micrometrics ASAP 2020 instrument (Micromeritics Instrument Co., Norcross, USA). FT-IR measurements of the samples were recorded on a NEXUS870 instrument (Nicolet Co., Ltd., USA). The zeta potentials of the samples were measured by using a Zetasizer instrument (Malvern, UK). The XPS spectra of samples were obtained from a PHI5000 Versa Probe XPS instrument (ULVAC-PHI, Japan).

## 2.3. Phosphate adsorption kinetics

To investigate the kinetics of phosphate adsorption, 0.1 g of $La_2O_3(14.7)@PCMK-3$ was added into 400 ml phosphate solution ($KH_2PO_4$ solution) with different initial phosphate concentrations (5, 10 and 20 mg P/L). The solution pH was adjusted to 7.0 by 1.0 M NaOH solution. Samples were collected at determined time intervals. After filtration using 0.45 µm membrane filter, the residual phosphate concentration of the filtered solution was analysed by the ammonium molybdate spectrophotometric method [32], which is a rapid, precise and widely used method for phosphate measurement. Phosphate adsorption amount on $La_2O_3(14.7)@PCMK-3$ was calculated as

$$q_t = \frac{(C_0 - C_t)V}{m},$$ (2.1)

where $C_0$ (mg l$^{-1}$) and $C_t$ (mg l$^{-1}$) are the initial phosphate concentration and phosphate concentration at time $t$, respectively; $m$ (g) is the mass of the adsorbent added and $V$ (l) is the phosphate solution volume.

## 2.4. Adsorption isotherms of phosphate

Phosphate adsorption isotherms were determined in duplicate in 40 ml glass vials with polytetrafluoroethylene-lined screw caps. Typically, 0.01 g of $La_2O_3(14.7)@PCMK-3$ was added into glass vials containing various initial concentrations of phosphate solution (1.5–20 mg P/L). Before adsorption experiments, solution pH was regulated with 0.1 M NaOH solution with an initial pH of approximately 7.0. Subsequently, the glass vials were transferred to a shaker and shaken for 48 h. The phosphate adsorption amount at equilibrium was defined as follows:

$$q_e = \frac{(C_0 - C_e)V}{m},$$ (2.2)

where $q_e$ is the equilibrium adsorption amount (mg g$^{-1}$); $C_0$ (mg l$^{-1}$) and $C_e$ (mg l$^{-1}$) is the initial concentration of phosphate and the equilibrium phosphate concentration; $m$ (g) is the mass of adsorbent and $V$ (l) is the volume of phosphate solution.

## 2.5. Influence of pH and coexisting anions

The effect of pH on the adsorption of phosphate in water was investigated in bath experiments. Phosphate solution (10 mg l$^{-1}$) and 0.01 g of La$_2$O$_3$(14.7)@PCMK-3 were added in glass vials, of which the initial pH values were adjusted ranging from 3.0 to 12.0 using 0.1 M NaOH or HCl. Then, the vials were shaken for 48 h. The final pH values of samples were determined and the adsorption amounts of phosphate at different pH values were measured by the method described above.

Besides pH influence, the adsorption selectivity for a given adsorbent is important prior to its practical application. Therefore, four common anions in natural water (Cl$^-$, SO$_4^{2-}$, NO$_3^-$ and F$^-$) were selected in this study to examine the phosphate adsorption selectivity of the adsorbent. The adsorbent (La$_2$O$_3$(14.7)@PCMK-3) was added to glass vials with 10 mg P/L of phosphate solution. The concentrations of the coexisting Cl$^-$, SO$_4^{2-}$, NO$_3^-$ and F$^-$ in batch experiments varied from 50 to 400 mg l$^{-1}$. Finally, the initial solution pH was adjusted to 7.0. The residual phosphate concentrations were determined after 48 h.

## 2.6. Desorption and regeneration

The regeneration of the adsorbent was conducted within five cycles to verify its reusability. Within each cycle, 0.01 g of La$_2$O$_3$(14.7)@PCMK-3 was dispersed in 10 mg l$^{-1}$ of phosphate solution (40 ml) under shaking. After reaching adsorption equilibrium, the phosphate adsorption amount by the adsorbent was measured. The used adsorbent was obtained by filtration and regenerated by mixing with 1 M NaOH solution under shaking for 24 h. Before the regeneration cycle, the adsorbent was washed with deionized water to neutral pH and dried at 60°C overnight.

# 3. Results and discussion

## 3.1. Material characterization

The SEM images of PCMK-3 and La$_2$O$_3$(14.7)@PCMK-3 are illustrated in electronic supplementary material, figure S1. It could be observed that the morphology of PCMK-3 was well maintained after lanthanum oxide loading. In the small-angle XRD spectra of SBA-15 (see figure 1*a*), sharp diffraction peaks at 0.84° (100) and three weak diffraction peaks at 2θ = 1.43° (110), 1.64° (200) and 2.16° (210) were observed, indicating the well-defined SBA-15 structure with hexagonal p6 mm symmetry [33–35]. Similarly, diffractions (100) and (200) were also identified on PCMK-3 and OCMK-3, indicating the retaining of the mesoporous structure after the replication process. Additionally, for OCMK-3, nitric acid oxidization did not influence the structure of PCMK-3. For La$_2$O$_3$ loaded PCMK-3, it could be clearly observed that the diffraction peaks of PCMK-3 decreased with the increase of La$_2$O$_3$ content (figure 1*b,c*), probably owing to contrast matching between La$_2$O$_3$ and carbon framework as a result of La$_2$O$_3$ loading into mesoporous carbon matrix [36,37]. The XRD patterns of La$_2$O$_3$(14.7)@PCMK-3 adsorbent after phosphate adsorption were analysed and displayed in the electronic supplementary material, figure S2. Compared with La$_2$O$_3$(14.7)@PCMK-3 before phosphate adsorption, new diffraction peaks were observed at 2θ = 19.9°, 31.0° and 41.8° from the pattern of La$_2$O$_3$(14.7)@PCMK-3 after phosphate adsorption, which were corresponding to the peaks of LaPO$_4$/LaPO$_4$·0.5H$_2$O crystal (JCPDS No. 75-1881/46-1439) [38], indicating that phosphate was adsorbed on La$_2$O$_3$(14.7)@PCMK-3. The TEM images of PCMK-3 and La$_2$O$_3$(14.7)@PCMK-3 are displayed in the electronic supplementary material, figure S3. The ordered structure of PCMK-3 and La$_2$O$_3$(14.7)@PCMK-3 was clearly visible, indicating the retaining of the mesoporous structure of PCMK-3 after La$_2$O$_3$ loading. For La$_2$O$_3$(14.7)@PCMK-3, the La$_2$O$_3$ particles in pores of PCMK-3 were invisible, indicating that the La$_2$O$_3$ particles were mainly presented in the form of an amorphous colloid. In order to confirm the distribution of La in the adsorbent, TEM-EDX analysis of La$_2$O$_3$(14.7)@PCMK-3 after adsorption was conducted, and the result is illustrated in figure 2. The corresponding EDX mapping showed La was evenly distributed in La$_2$O$_3$(14.7)@PCMK-3 (figure 2*d*), confirming La species were effectively confined in PCMK-3 mesopores. In addition, P elements are perfectly coordinating to the distribution of La elements in figure 2*e*, indicating that La was the active species responsible for phosphate adsorption. To further clarify the composition change of La$_2$O$_3$(14.7)@PCMK-3 before and after adsorption, the high-resolution XPS spectra of La$_2$O$_3$(14.7)@PCMK-3 are displayed in the electronic supplementary material, figure S4. For the high-resolution spectra of La 3d before adsorption (electronic supplementary material,

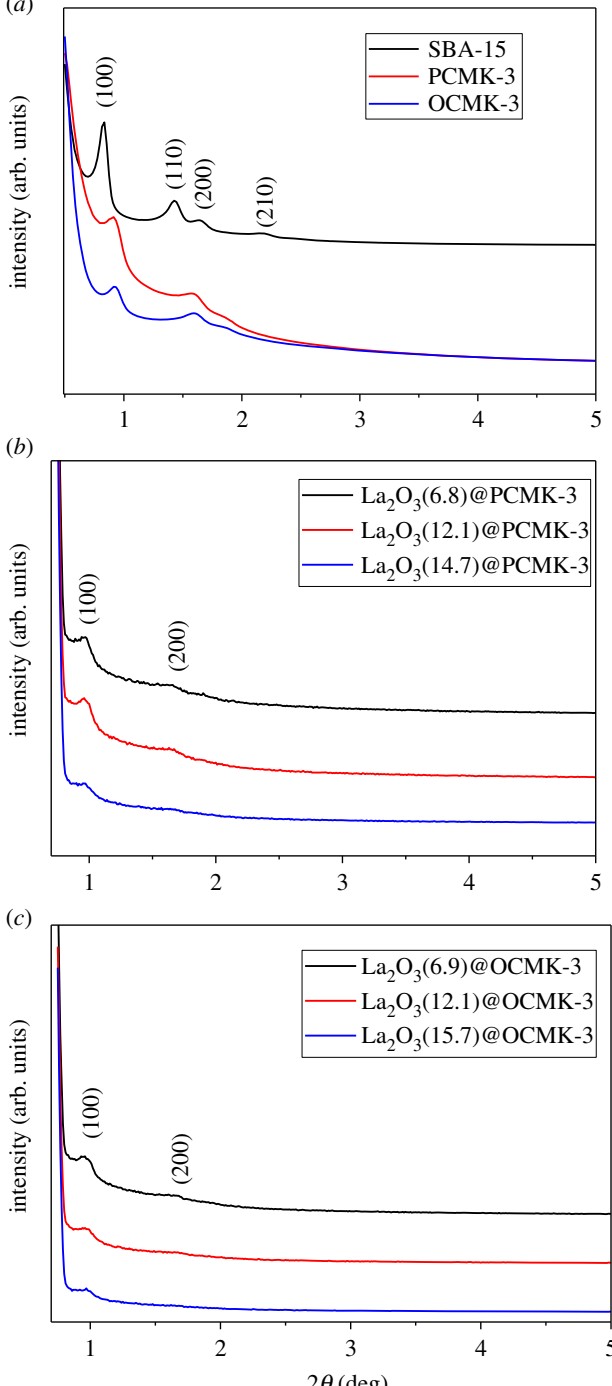

**Figure 1.** Small-angle XRD patterns of the samples.

figure S4*a*), the peaks at 835.75 eV and 852.38 eV could be ascribed to $La_2O_3$, and the peaks at 838.80 eV and 855.63 eV were corresponding to $La(OH)_3$ [39]. After phosphate adsorption, a slight positive binding energy shift (approx. 0.625 eV) of $La_2O_3$ was determined, indicating the formation of $LaPO_4$ complex [40]. For P 2p scanning spectra in the electronic supplementary material, figure S4*b*, the peak at a binding energy of 132.8 eV corresponding to P 2p was clearly identified after phosphate adsorption, indicating the formation of $LaPO_4$ compound [41].

The surface zeta potential of the adsorbent was a key parameter correlated to its adsorption performance. Hence, zeta potentials of the samples were tested and the relationship of zeta potential versus pH is presented in figure 3. Due to deprotonation of surface hydroxide groups, zeta potentials of all samples presented a monotonous decrease with pH increasing. It should also be noted that the

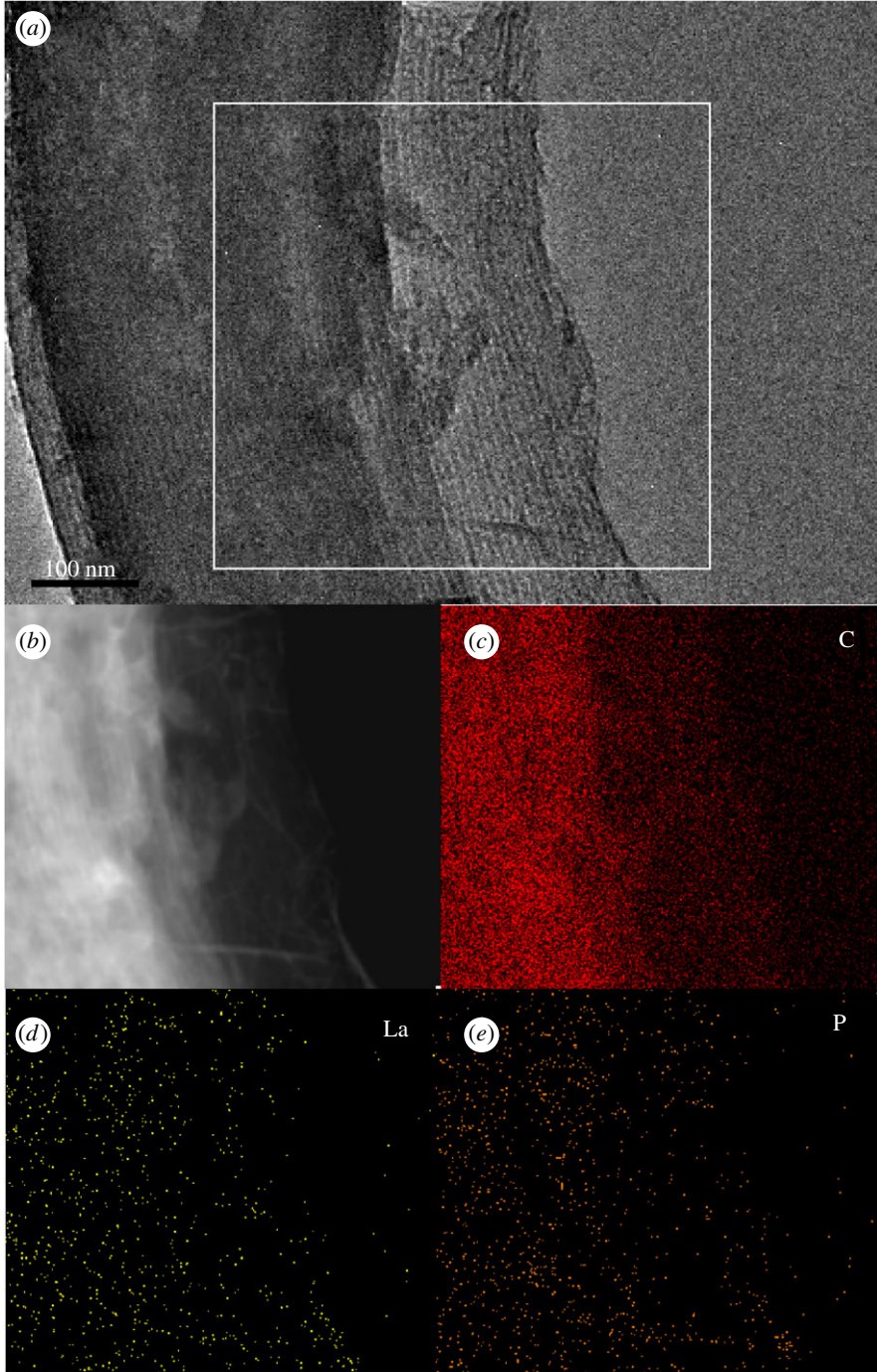

**Figure 2.** The TEM image of La$_2$O$_3$(14.7)@PCMK-3 after phosphate adsorption (*a*). The EDX mapping selected area (*b*) and the EDX images of C (*c*), La (*d*) and P (*f*).

zeta potential of OCMK-3 in the test pH range was lower than that of PCMK-3, probably attributed to the formation of carboxyl groups in OCMK-3 after nitric acid oxidation [42]. This is supported by the FT-IR spectra of PCMK-3 and OCMK-3 samples (figure 4). The spectra of all PCMK-3 and OCMK-3 samples showed the bands at 1565 cm$^{-1}$ and 1130 cm$^{-1}$, which were assigned to C=C stretching vibrations and skeletal C-C tangential motions [43], respectively. Compared with PCMK-3 samples, a weak band at 1715 cm$^{-1}$ could be observed in the FT-IR spectrum of OCMK-3 samples (figure 4*b*), which is assigned to C=O stretching vibrations from oxygen-containing groups on the carbon surface, such as carboxylic acids. A distinct band was observed at 577 cm$^{-1}$, indicating the presence of La–OH bond vibration [18]. The zeta potentials of La$_2$O$_3$@PCMK-3 and La$_2$O$_3$@OCMK-3 were higher than those of PCMK-3 and OCMK-3, probably owing to the loading of La$_2$O$_3$. The FT-IR spectra of

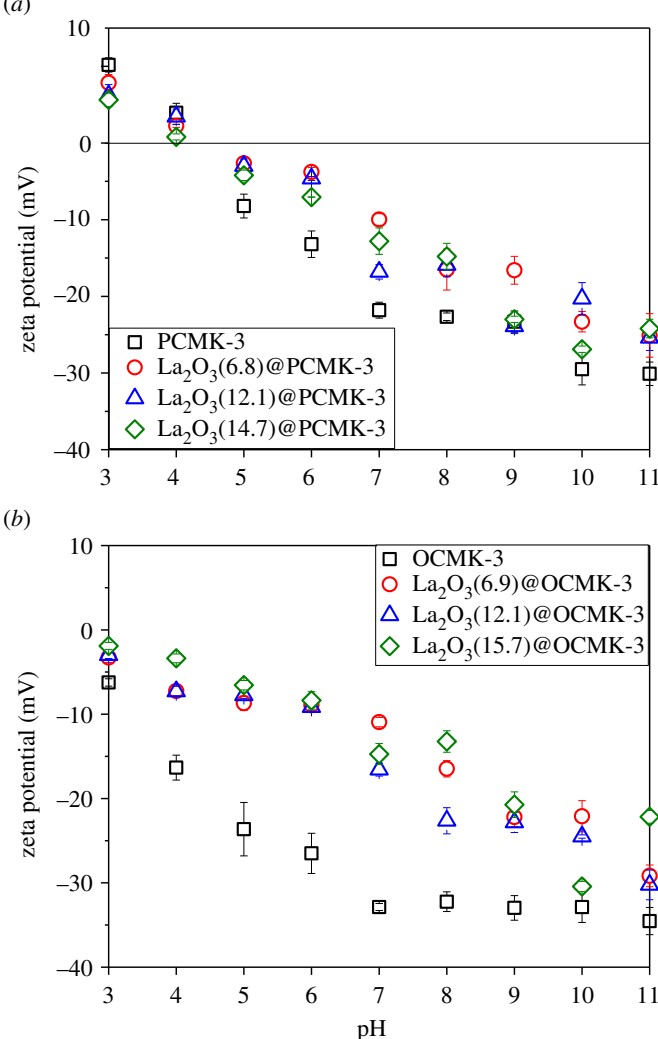

**Figure 3.** Zeta potentials of the samples.

$La_2O_3(14.7)@PCMK\text{-}3$ before and after adsorption is displayed in the electronic supplementary material, figure S5. After phosphate adsorption, a strong absorption peak was clearly identified at about $1054\ cm^{-1}$, which could be assigned to the stretching vibration of P-O of $PO_4^{3-}$ group [18,44]. In addition, the peaks at $613\ cm^{-1}$ and $541\ cm^{-1}$ should be attributed to the bend vibration of O-P-O [45], confirming that phosphate was adsorbed by $La_2O_3(14.7)@PCMK\text{-}3$.

Figure 5 and table 1 illustrate $N_2$ adsorption–desorption isotherms and pore parameters of the samples, respectively. In figure 5, all samples had type IV isotherm consisting of an H1 hysteresis loop in $N_2$ adsorption–desorption curves, suggesting the presence of mesopores [46]. Notably, the isotherms of adsorbents showed a decrease of $N_2$ adsorption amount with increasing $La_2O_3$ loading amount, likely due to $La_2O_3$ loading in the mesopores of PCMK-3 and OCMK-3. Consistently, the BET surface areas were 915.9, 879.3 and 866.9 $m^2/g$ for the $La_2O_3@PCMK\text{-}3$ samples with $La_2O_3$ loadings of 6.8, 12.1 and 14.7 wt.%, respectively, indicating that the increase of $La_2O_3$ loading led to the decreased surface area of the adsorbents. A similar trend was also observed on $La_2O_3@OCMK\text{-}3$. Accordingly, the mesopore volumes of the samples with varied $La_2O_3$ contents were calculated, and the results showed that $La_2O_3$ loading led to distinct decreases of mesopore volumes of PCMK-3 and OCMK-3. Assuming that $La_2O_3$ is only deposited on the external surface of PCMK-3, the mesoporous volume of the composite can be calculated as $V_{Total} = M_{La_2O_3} \times V_{La_2O_3} + M_{P(O)CMK\text{-}3} \times V_{P(O)CMK\text{-}3}$, where $M_{La2O3}$ and $M_{P(O)CMK\text{-}3}$ are the mass per cent of $La_2O_3$ and P(O)CMK-3, respectively; $V_{La2O3}$ and $V_{P(O)CMK\text{-}3}$ are the mesopore volume of $La_2O_3$ and P(O)CMK-3, respectively. From table 2, therefore, it could be easily concluded that the tested mesopore volumes of $La_2O_3@PCMK\text{-}3$ and $La_2O_3@OCMK\text{-}3$ are less than those of calculated values, reflecting the effective confinement of $La_2O_3$ in PCMK-3 pores. To further clarify the confinement effects, electronic supplementary material, figure S6 exhibits

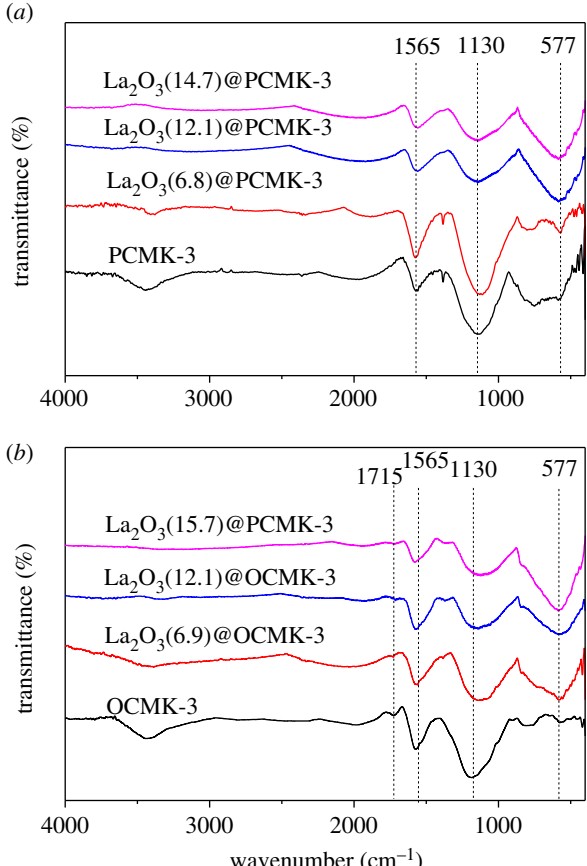

**Figure 4.** IR spectra of PMCK-3 samples (*a*) and OMCK-3 samples (*b*).

the relationship of pore volume filling percentages and $La_2O_3$ loading amounts of samples. A linear relationship could be clearly observed between pore volume filling percentage and $La_2O_3$ loading amount, which indicated that $La_2O_3$ particles were well dispersed in pores of P(O)CMK-3. In addition, compared with $La_2O_3$@OCMK-3, the curve of $La_2O_3$@PCMK-3 in the electronic supplementary material, figure S7 became flat at $La_2O_3$ loading amount above 12.1 wt.%. This difference indicated that when increasing the $La_2O_3$ loading amount, the hydrophilic surface of OCMK-3 was more favourable for the effective dispersion of $La_2O_3$ particles. In conclusion, the results confirmed that the $La_2O_3$ particles were mainly confined in the pore other than on the external surface of carriers. For $La_2O_3$@PCMK-3 and $La_2O_3$@OCMK-3, the pore diameters were identified to be around 4 nm (see electronic supplementary material, figure S3), which were identical to the pore diameters of PCMK-3 and OCMK-3. However, compared with PCMK-3 and OCMK-3, the peak intensity of the pore volume distribution of adsorbents decreased with the increasing of $La_2O_3$ loading amount, again reflecting that the mesopores were effectively occupied by $La_2O_3$ particles.

## 3.2. Phosphate adsorption kinetics

The rate of phosphate removal by the adsorbent from an aqueous solution was examined in batch kinetic experiments. The kinetics of phosphate adsorption onto $La_2O_3$(14.7)@PCMK-3 with different initial phosphate concentrations are presented in figure 6. The adsorption of phosphate with varied initial concentrations displayed similar adsorption kinetics. Very fast adsorption appeared in the initial adsorption stage, and phosphate was adsorbed by 80% within 240 min. Additionally, phosphate adsorption capacity increased with initial phosphate concentration. The kinetics data were further fitted to two kinetic models, e.g. the pseudo-first-order and pseudo-second-order adsorption models. In terms of correlation coefficient values ($R^2 > 0.99$), the pseudo-second-order model (equation 3) was more suitable for describing phosphate adsorption on $La_2O_3$(14.7)@PCMK-3.

$$\frac{t}{q_t} = \frac{1}{k_2 q_e^2} + \frac{1}{q_e} t,$$

(3.1)

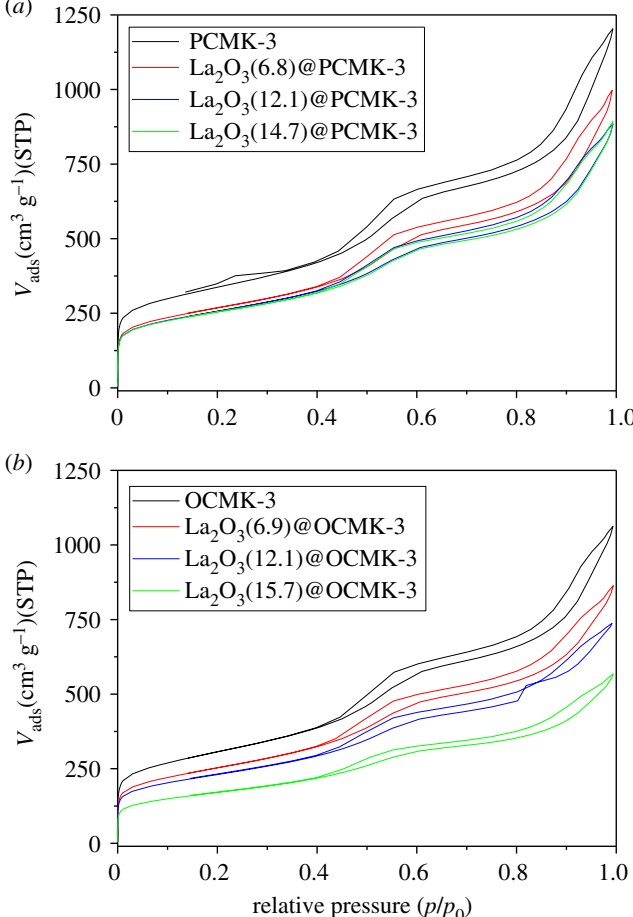

**Figure 5.** $N_2$ adsorption–desorption isotherms of the samples.

**Table 1.** Structural parameters of PCMK-3, OCMK-3, La$_2$O$_3$@PCMK-3 and La$_2$O$_3$@OCMK-3.

| samples | $S_{BET}$ (m$^2$/g) | $V_t^a$ (cm$^3$/g) | $V_{meso}^b$ (cm$^3$/g) | $V_{micro}^c$ (cm$^3$/g) | $D_{mp}^d$ (nm) |
|---|---|---|---|---|---|
| PCMK-3 | 1148.5 | 1.73 | 1.58 | 0.15 | 3.89 |
| La$_2$O$_3$(6.8)@PCMK-3 | 915.9 | 1.38 | 1.28 | 0.10 | 3.91 |
| La$_2$O$_3$(12.1)@PCMK-3 | 879.3 | 1.24 | 1.14 | 0.10 | 3.94 |
| La$_2$O$_3$(14.7)@PCMK-3 | 866.7 | 1.23 | 1.13 | 0.10 | 3.98 |
| OCMK-3 | 1048.8 | 1.52 | 1.41 | 0.11 | 3.94 |
| La$_2$O$_3$(6.9)@OCMK-3 | 869.0 | 1.23 | 1.16 | 0.07 | 3.85 |
| La$_2$O$_3$(12.1)@OCMK-3 | 790.6 | 1.08 | 1.00 | 0.08 | 3.97 |
| La$_2$O$_3$(15.7)@OCMK-3 | 585.4 | 0.81 | 0.75 | 0.06 | 4.02 |

$^a$total pore volume, at $p/p_0 = 0.98$.
$^b$mesopore volume, calculated by $V_t − V_{micro}$.
$^c$micropore volume.
$^d$most probable pore diameter, calculated using the BJH method.

where $q_t$ (mg g$^{-1}$) and $q_e$ (mg g$^{-1}$) are the phosphate uptakes at time $t$ (min) and at equilibrium, respectively; $k_2$ (g/(mg·min)) is the rate constant of pseudo-second-order kinetics.

The fitting results showed that the $k_2$ values were $1.5 \times 10^{-3}$, $1.2 \times 10^{-3}$ and $1.2 \times 10^{-3}$ g/(mg min) for phosphate adsorption on La$_2$O$_3$(14.7)@PCMK-3 with initial phosphate concentrations of 5, 10 and 20 mg l$^{-1}$, respectively. The fitting result from electronic supplementary material, table S1 indicated that the adsorption rates of phosphate on La$_2$O$_3$(14.7)@PCMK-3 were approximately the same within

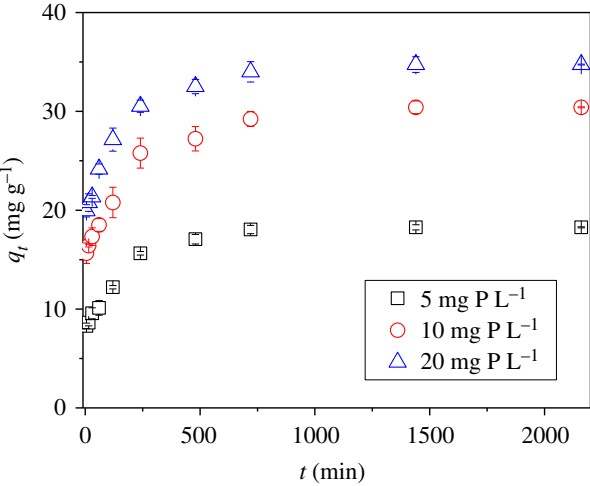

**Figure 6.** Adsorption kinetics of phosphate onto La$_2$O$_3$(14.7)@PCMK-3 with initial phosphate concentration of 5, 10 and 20 mg P/L.

**Table 2.** Isotherm parameters for phosphate adsorption on the adsorbents fitted by the Freundlich model.

| samples | $1/n$ | $K_F$ (mg P$^{1-n}$/(L$^n$ · g)) | $R^2$ |
|---|---|---|---|
| La$_2$O$_3$(6.8)@PCMK-3 | 0.25 | 6.84 | 0.94 |
| La$_2$O$_3$(12.1)@PCMK-3 | 0.14 | 15.32 | 0.83 |
| La$_2$O$_3$(14.7)@PCMK-3 | 0.18 | 21.23 | 0.89 |
| La$_2$O$_3$(6.9)@OPCMK-3 | 0.22 | 9.50 | 0.88 |
| La$_2$O$_3$(12.1)@OCMK-3 | 0.12 | 19.97 | 0.98 |
| La$_2$O$_3$(15.7)@OCMK-3 | 0.16 | 27.25 | 0.93 |

a broad initial phosphate concentration range. The almost identical adsorption kinetics is ascribed for the open three-dimensional framework of La$_2$O$_3$(14.7)@PCMK-3 with highly accessible adsorption sites, which are favourable for phosphate to feasibly overcome the diffusion resistance.

## 3.3. Phosphate adsorption isotherms

The isotherms of phosphate adsorption on the adsorbents are compared in figure 7. It was clear that phosphate adsorption capacities of La$_2$O$_3$@PCMK-3 and La$_2$O$_3$@OCMK-3 increased with the La$_2$O$_3$ loading content, respectively, indicating that La$_2$O$_3$ was the active site responsible for the adsorption of phosphate. Notably, in the low phosphate concentration range, the adsorbents showed excellent phosphate removal efficiency. For example, at an initial phosphate concentration of 1.5 mg l$^{-1}$, phosphate adsorption on La$_2$O$_3$(14.7)@PCMK-3 and La$_2$O$_3$(15.7)@OCMK-3 was higher than 97%. The result indicated that it was suitable for La$_2$O$_3$@PCMK-3 and La$_2$O$_3$@OCMK-3 to be applied in the wastewater with low phosphate concentrations. In addition, at an approximately equal loading amount of La$_2$O$_3$, La$_2$O$_3$@OCMK-3 adsorbents exhibited higher phosphate adsorption capacities than those of La$_2$O$_3$@PCMK-3 adsorbents. The main reason could be ascribed to the presence of abundant oxygen-containing functional groups formed after nitric acid treating, which increased the surface hydrophilicity of PCMK-3 and hence the adsorption to hydrophilic phosphate. Moreover, the presence of hydrophilic groups in OCMK-3 possibly enhanced the dispersion of La$_2$O$_3$ species and increased the amount of explored La$_2$O$_3$ for phosphate adsorption.

To further clarify the phosphate adsorption mechanism, the experimental data were fitted to two classical isotherm models (i.e. Langmuir and Freundlich isotherms) which were expressed in equations (3.2) and (3.2):

$$\text{Langmuir equation} \quad q_e = q_m K_L \frac{C_e}{1 + K_L C_e} \tag{3.2}$$

$$\text{Freundlich equation} \quad q_e = K_F C_e^{1/n} \tag{3.3}$$

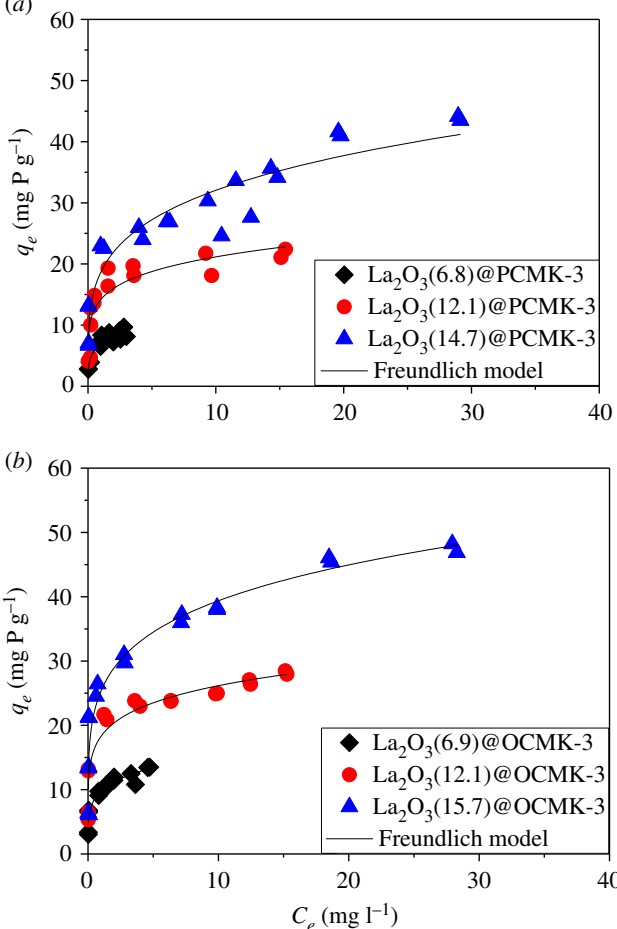

**Figure 7.** Isotherms of phosphate adsorption on La$_2$O$_3$@PCMK-3 (*a*) and La$_2$O$_3$@OCMK-3 (*b*).

where $q_e$ (mg g$^{-1}$) is the adsorption amount at an equilibrium concentration of $C_e$ (mg l$^{-1}$); $q_m$ (mg g$^{-1}$) is the maximum capacity; $K_L$ (l mg$^{-1}$) is the Langmuir adsorption constant. $K_F$ (mg P$^{1-n}$/(L$^n$ g)) and $n$ are the Freundlich affinity coefficient and linearity constant, respectively.

Table 2 lists the parameters fitted by the Freundlich model, which could describe phosphate adsorption on the adsorbents well. This indicates the presence of heterogeneous adsorption sites from the adsorbents. As shown in table 2, the $n$ values of phosphate adsorption on the samples were larger than one, reflecting a favourable adsorption affinity of phosphate towards adsorbents. With approximately equal La$_2$O$_3$ loading contents, the $K_F$ values were calculated to be 15.32 and 19.97 mg P$^{1-n}$/(L$^n$ g) for La$_2$O$_3$(12.1)@PCMK-3 and La$_2$O$_3$(12.1)@OCMK-3, respectively, again confirming a phosphate adsorption affinity order of La$_2$O$_3$@OCMK-3 > La$_2$O$_3$@PCMK-3. Furthermore, compared with other La-based adsorbents reported in the electronic supplementary material, table S2, La$_2$O$_3$(14.7)@PCMK-3 and La$_2$O$_3$(15.7)@OCMK-3 showed considerable phosphate adsorption capacity.

The previous studies reported that confinement was favourable for the dispersion of active sites without aggregation, further enhancing the reactivity of active sites significantly [47,48]. For the samples with different La$_2$O$_3$ loading contents, the adsorption isotherms normalized by La$_2$O$_3$ loading content can offer a clearer insight into how the confinement effect influences the adsorption performances of the adsorbents. Figure 8 plots normalized $q_e$ defined as adsorption amount divided by mass per cent of La$_2$O$_3$ (wt.%) versus phosphate concentration ($C_e$) at equilibrium. Despite different La$_2$O$_3$ loading, the La$_2$O$_3$ normalized phosphate adsorption capacities of La$_2$O$_3$@PCMK-3 adsorbents showed a comparable trend within tested phosphate concentrations (see figure 8). Similarly, the La$_2$O$_3$@OCMK-3 adsorbents showed the same result. Furthermore, compared with La$_2$O$_3$@PCMK-3 and La$_2$O$_3$@OCMK-3 adsorbents, pure La$_2$O$_3$ exhibited much lower phosphate adsorption capacity normalized by La$_2$O$_3$ loading, probably ascribed to the formation and/or aggregation of large La$_2$O$_3$ particles. The results clearly indicated that confinement of La$_2$O$_3$ particles effectively inhibited particle aggregation in PCMK-3 and

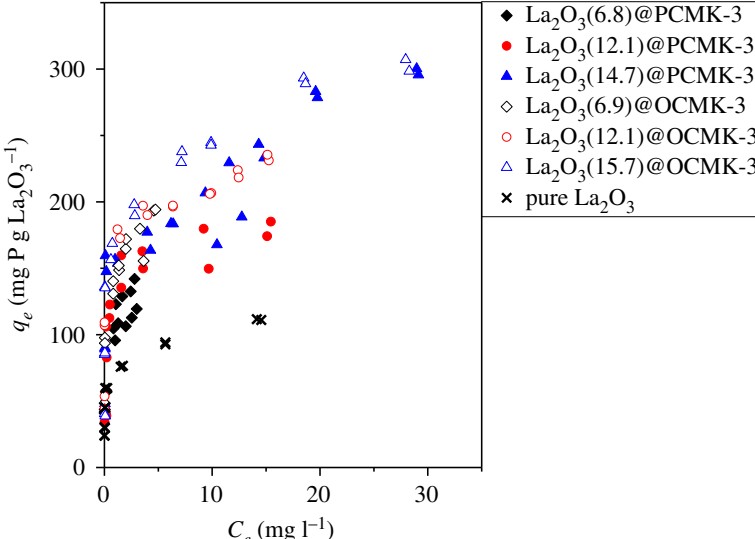

**Figure 8.** Phosphate adsorption isotherms of the adsorbents normalized by $La_2O_3$ content.

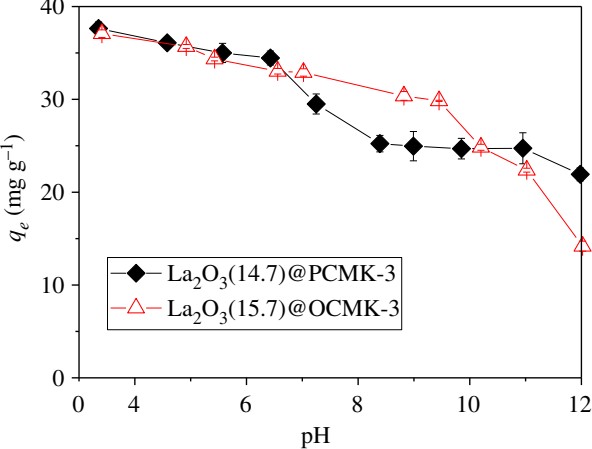

**Figure 9.** Influence of pH on phosphate adsorption on $La_2O_3$(14.7)@PCMK-3 and $La_2O_3$(15.7)@OCMK-3.

OCMK-3 pores, and the solid-state grinding method was an effective method to confine $La_2O_3$ particles in PCMK-3 and OCMK-3 in a wide $La_2O_3$ loading range.

## 3.4. The influence of pH and coexisting anions on phosphate adsorption

Water chemistry of the aqueous solution, such as pH and coexisting anions, is commonly considered to be an important factor that affects phosphate adsorption amounts of the adsorbents [16,49–51]. Figure 9 illustrates the influence of pH on phosphate adsorption on $La_2O_3$(14.7)@PCMK-3 and $La_2O_3$(15.7)@OCMK-3. Within the test pH range, phosphate adsorption was found to be strongly dependent on the final pH of the solution. The adsorption capacities of $La_2O_3$(14.7)@PCMK-3 and $La_2O_3$(15.7)@OCMK-3 were measured to be 37.64 mg g$^{-1}$ and 37.08 mg g$^{-1}$ at pH = 3.4. Increasing pH to 12.0, a significant decrease of adsorption amount to 21.92 mg g$^{-1}$ and 14.18 mg g$^{-1}$ was observed, reflecting a strong pH dependence of phosphate adsorption on $La_2O_3$(14.7)@PCMK-3 and $La_2O_3$(15.7)@OCMK-3. The result was consistent with previous reports on other La-based adsorbents [18,52,53].

The main mechanism of phosphate adsorption on $La_2O_3$(14.7)@PCMK-3 could be ascribed to the formation of the inner-sphere complex between phosphate and $La_2O_3$ [21]. The hydroxyls from La-OH groups of adsorbents could be exchanged by phosphate. Given p$K_a$ values of 2.1, 7.2 and 12.3 in water, phosphate mainly exists in the forms of $HPO_4^{2-}$ or $H_2PO_4^-$ in the pH range of 3–12. Within the low pH range, the La-OH groups from $La_2O_3$(14.7)@PCMK-3 and $La_2O_3$(15.7)@OCMK-3 were protonated and positively charged, which was favourable for attracting anionic phosphate.

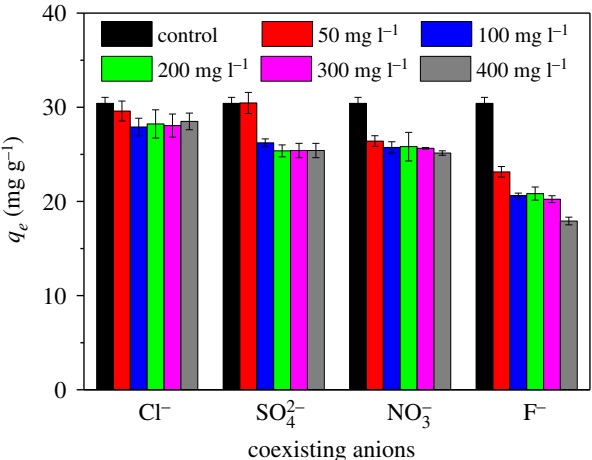

**Figure 10.** The effect of chloride, sulfate, nitrate and fluoride (50–400 mg l$^{-1}$) on phosphate adsorption.

Consequently, as the pH value became lower, La-OH groups became more protonated and this enhanced the phosphate displacement. When pH increased, the active sites were gradually negatively charged via deprotonation, which had an electrostatic repulsive force with anionic phosphate. Furthermore, OH$^-$ concentration increased with pH and competed with anionic phosphate for adsorption sites, further suppressing phosphate adsorption at high pH. Accordingly, zeta potential results confirmed the monotonic decrease of zeta potentials with pH in the range of 3–11 because of the deprotonation of the surface of La$_2$O$_3$(14.7)@PCMK-3 and La$_2$O$_3$(15.7)@OCMK-3 (see figure 3). With pH increasing from 3 to 11, the surface charge of adsorbents became more negative and then the repulsive force towards phosphate dominated. Notably, with pH increasing from 3 to 11, the zeta potentials of La$_2$O$_3$(14.7)@PCMK-3 and La$_2$O$_3$(15.7)@OCMK-3 decreased from 5.63 mV to −24.2 mV and −1.9 mV to −22.17 mV, respectively. Therefore, at high pH, the negatively charged surface was expected to inhibit the adsorption of phosphate on active sites, which led to continuously decreased phosphate adsorption with the final pH of the solution.

The impacts of coexisting anions (Cl$^-$, SO$_4^{2-}$, NO$_3^-$ and F$^-$) were investigated and the results are depicted in figure 10. It could be clearly observed that even at an anion concentration of 400 mg l$^{-1}$, phosphate adsorption on La$_2$O$_3$(14.7)@PCMK-3 was only slightly influenced by Cl$^-$, SO$_4^{2-}$ and NO$_3^-$ anions. For example, the phosphate adsorption capacities decreased only by 6–17% when the concentrations of Cl$^-$, SO$_4^{2-}$ or NO$_3^-$ increased to 400 mg l$^{-1}$. In principle, Cl$^-$, NO$_3^-$ and SO$_4^{2-}$ anions probably competed for the adsorption sites via the electrostatic interaction. The very low adsorption competitiveness of the coexisting anions can be explained by the fact that phosphate was adsorbed on La$_2$O$_3$ primarily via the formation of highly stable inner-sphere complexes [21]. The schematic representation of the mechanism for phosphate adsorption on La$_2$O$_3$(14.7)@PCMK-3 is illustrated in the electronic supplementary material, figure S8. As a result, the presence of these ubiquitous anions did not remarkably influence phosphate adsorption even at very high concentrations. By contrast, within the test concentration range, F$^-$ anion displayed a negative impact on phosphate adsorption. Increasing F$^-$ concentration to 400 mg l$^{-1}$, the adsorption capacity of La$_2$O$_3$(14.7)@PCMK-3 for phosphate decreased from 30.41 mg g$^{-1}$ to 19.72 mg g$^{-1}$. The significantly suppressed phosphate adsorption by F$^-$ may be attributed to the very small $K_{sp}$ (less than 10$^{-26}$) of LaF$_3$ as well as the strong complexing capability of F$^-$ with La$_2$O$_3$ [54]. Similar inhibition effects were also observed previously [18]. Generally, La$_2$O$_3$(14.7)@PCMK-3 exhibited highly selective phosphate adsorption performance.

## 3.5. Desorption and regeneration

The re-use of the adsorbent by adsorption and regeneration was tested to examine the reusability of the adsorbent within five cycles. The used adsorbent was regenerated under alkaline conditions and the result is presented in figure 11. For the first adsorption–desorption cycle, phosphate adsorption capacity of La$_2$O$_3$(14.7)@PCMK-3 decreased from 29.78 m/g to 18.21 mg g$^{-1}$, probably because of partial loss of La$_2$O$_3$ particles located on the external region of PCMK-3 or the presence of very strong phosphate adsorption, which could not be regenerated during the regeneration process. As for the subsequent four adsorption–desorption cycles, however, the adsorption capacities of La$_2$O$_3$(14.7)@PCMK-3 remained

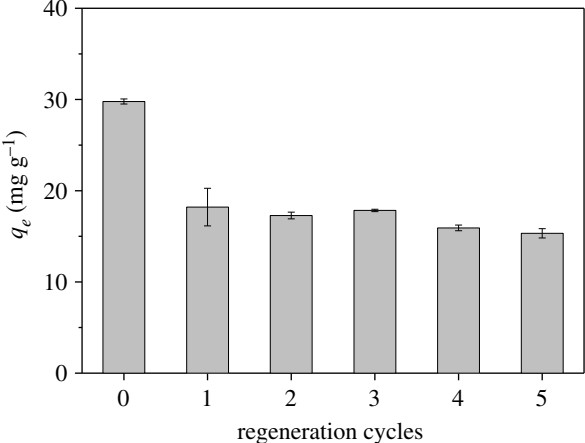

**Figure 11.** Re-use of $La_2O_3(14.7)$@PCMK-3 for phosphate adsorption.

almost constant, displaying highly stable adsorption–desorption performances. The results clearly showed that $La_2O_3(14.7)$@PCMK-3 could be repeatedly used for the removal of phosphate in water.

# 4. Conclusion

In this study, two La-confined adsorbents with PCMK-3 and OCMK-3 as carriers were prepared for phosphate adsorption. The results from adsorbent characterization indicate that increasing $La_2O_3$ loading content results in the decrease of the pore volume of PCMK-3 and OCMK-3 by 20.2%–28.9% and 19.1%–46.7%, respectively, reflecting that $La_2O_3$ particles are well confined in the pores of carbon carriers. For phosphate adsorption, the adsorption capacities of $La_2O_3$@PCMK-3 and $La_2O_3$@OCMK-3 increase with $La_2O_3$ loading amount. Furthermore, $La_2O_3$@PCMK-3 and $La_2O_3$@OCMK-3 adsorbents with various $La_2O_3$ loading contents exhibit approximately identical normalized adsorption capacities. The result demonstrates that the confinement effect is highly effective within a broad $La_2O_3$ loading amount range. The kinetics of phosphate adsorption on $La_2O_3(14.7)$@PCMK-3 can be well described by pseudo-second-order kinetic model. The phosphate adsorption amount was strongly dependent on solution pH and markedly decreased with pH increasing. Coexisting anions ($Cl^-$, $NO_3^-$, and $SO_4^{2-}$) have a negligible influence on phosphate adsorption, indicative of a highly selective adsorption affinity of $La_2O_3(14.7)$@PCMK-3 for phosphate. The present study indicates that $La_2O_3$ confined in mesoporous carbon is a promising adsorbent for phosphate removal.

Data accessibility. The datasets supporting this article have been uploaded as part of the electronic supplementary material.

Authors' contribution. X.J. and H.C. carried out the experiment, participated in data analysis and drafting the manuscript; T.L. designed the work, carried out data analysis, interpreted the results and helped draft the manuscript; Y.S. participated in interpreting the data and the preparation of the manuscript; S.Z. and X.Q. co-supervised the project and proofread the manuscript.

Competing Interests. The authors declare that they do not have any commercial or associative interest that could have appeared to influence the work reported in this paper.

Funding. The authors acknowledge the financial support from Research Program Foundation of Nanjing Institute of Technology (grant no. YKJ201935), the National Natural Science Foundation of China (grant no. 21976086) and State Key Laboratory of Pollution Control and Resource Reuse open fund (grant no. PCRRF20014).

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
