## [Peer Review File · Royal Society Open Science]

Review History

RSOS-210428.R0 (Original submission)

Review form: Reviewer 1

Is the manuscript scientifically sound in its present form?

Yes

Are the interpretations and conclusions justified by the results?

Yes

Is the language acceptable?

Yes

Do you have any ethical concerns with this paper?

No

Have you any concerns about statistical analyses in this paper?

No

Recommendation?

Accept with minor revision (please list in comments)

Comments to the Author(s)

This work reports the encapsulation of lanthanum oxide particles inside pristine and oxidized mesoporous carbon particles by solid state grinding method. The resulting La₂O₃@PCMK-3 and La₂O₃@OCMK-3 composites have been characterized by different techniques, such as XRD, TEM, BET, FTIR, XRF, Beta-potential, etc. These composites have been tested as adsorbents for phosphate. They found that La₂O₃@OCMK-3 samples exhibited the best performance for phosphate adsorption with high adsorption capacity and good regeneration ability. Overall, the work is interesting and explained well. It can be accepted after the following minor revisions:

1. The FT-IR spectra of La₂O₃(14.7)@PCMK-3 before and after adsorption should be provided and analyzed.
2. How about the morphology of La₂O₃(14.7)@PCMK-3 after phosphate adsorption? Is the mesoporous structure still maintained?
3. The IR spectra of all La₂O₃@PMCK-3 and La₂O₃@OMCK-3 should be presented rather than general IR spectra of PMCK-3 and OMCK-3, since other characterization data present all the samples.
4. Phosphate adsorption performance of the optimum catalyst, La₂O₃(14.7)@PCMK-3, should be compared with other adsorbents reported in the literature.
5. The purity of the NH₃ atmosphere used to prepare La-confined PCMK-3 should be specified in the Experimental Section.
6. The references are a little outdated. More recent references on template-based fabrication of mesoporous carbon materials, such as Chemical Society Reviews 49, 4681-4736 (2020); Microporous Mesoporous Mater., 307, 110510 (2020); Chemical Engineering Journal 396, 125154 (2020); Journal of Materials Science, 56, 3312-3324 (2021); Electrochimica Acta, 332, 135399 (2020) are recommended to be cited in the Introduction or other appropriate parts.

Review form: Reviewer 2

Is the manuscript scientifically sound in its present form?

No

Are the interpretations and conclusions justified by the results?

No

Is the language acceptable?

No

Do you have any ethical concerns with this paper?

No

Have you any concerns about statistical analyses in this paper?

No

Recommendation?

Major revision is needed (please make suggestions in comments)

Comments to the Author(s)

In this paper, the phosphate adsorbents with confined La₂O₃ inside mesoporous carbon were prepared by the solid state grinding method using pristine mesoporous carbon material CMK-3 (PCMK-3) and oxidized CMK-3 (OCMK-3) as the matrix (denoted as La₂O₃@PCMK-3 and La₂O₃@OCMK-3), and their adsorption performances for phosphate were investigated. The La-

confined adsorbents prepared by the solid state grinding method were well characterized. I think this paper can be considered for publication after major revision. The detail comments are listed as follows:

- (1) It is suggested to assign and mark specific peaks on XRD curves.
- (2) How does the pH affect the adsorption capacity of La₂O₃@ocmk-3 adsorbent? Effect of pH should be discussed, and correlated with Zeta potential, and the pH of solution after adsorption.
- (3) The selectivity of La₂O₃@PCMK-3 is better, why? Is it related to the structure? Please give a detail explanation.
- (4) The R² values obtained by fitting with Freundlich model are all less than 0.99, especially for La₂O₃ (14.7)%@PCMK-3, which cannot strongly indicate that the adsorption behavior conforms to Freundlich model.
- (5) From Figure 7, it can be seen that the phosphate adsorption capacity of the adsorbent after normalization of La₂O₃ has great difference, which is basically the same as the adsorption capacity described in the paper.
- (6) From Fig. S3., when the loading amount of La₂O₃ in PCMK-3 is 14.7%, it still has an upward trend compared with the previous point. It is suggested to do two points to ensure that 14.7% is the best loading amounts.
- (7) Please improve language greatly.
- (8) SEM, TEM and corresponding Elemental mapping are necessary to confirm the morphology and the distribution of La₂O₃ on the matrix. Author must provide these data.
- (9) The XRD and XPS analysis of adsorbent after phosphate adsorption should be made to confirm the change of valence of metal element, and also the formation of new crystal phases.

Decision letter (RSOS-210428.R0)

Dear Dr Liu:

Title: Confined La₂O₃ particles in mesoporous carbon material for enhanced phosphate adsorption

Manuscript ID: RSOS-210428

The editor assigned to your manuscript has now received comments from reviewers. We would like you to revise your paper in accordance with the referee and Subject Editor suggestions which can be found below (not including confidential reports to the Editor). Please note this decision does not guarantee eventual acceptance.

Please submit your revised paper before 04-Jun-2021. Please note that the revision deadline will expire at 00.00am on this date. If we do not hear from you within this time then it will be assumed that the paper has been withdrawn. In exceptional circumstances, extensions may be possible if agreed with the Editorial Office in advance. We do not allow multiple rounds of revision so we urge you to make every effort to fully address all of the comments at this stage. If deemed necessary by the Editors, your manuscript will be sent back to one or more of the original reviewers for assessment. If the original reviewers are not available we may invite new reviewers.

On behalf of the Subject Editor Professor Anthony Stace and the Associate Editor Professor Chaohua Cui.

RSC Associate Editor:
Comments to the Author:
(There are no comments.)

RSC Subject Editor:
Comments to the Author:
(There are no comments.)

Reviewers' Comments to Author:

Reviewer: 1

Comments to the Author(s)

This work reports the encapsulation of lanthanum oxide particles inside pristine and oxidized mesoporous carbon particles by solid state grinding method. The resulting La₂O₃@PCMK-3 and La₂O₃@OCMK-3 composites have been characterized by different techniques, such as XRD, TEM, BET, FTIR, XRF, Beta-potential, etc. These composites have been tested as adsorbents for phosphate. They found that La₂O₃@OCMK-3 samples exhibited the best performance for phosphate adsorption with high adsorption capacity and good regeneration ability. Overall, the work is interesting and explained well. It can be accepted after the following minor revisions:

1. The FT-IR spectra of La₂O₃(14.7)@PCMK-3 before and after adsorption should be provided and analyzed.

2. How about the morphology of La₂O₃(14.7)@PCMCK-3 after phosphate adsorption? Is the mesoporous structure still maintained?
3. The IR spectra of all La₂O₃@PMCK-3 and La₂O₃@OMCK-3 should be presented rather than general IR spectra of PMCK-3 and OMCK-3, since other characterization data present all the samples.
4. Phosphate adsorption performance of the optimum catalyst, La₂O₃(14.7)@PCMCK-3, should be compared with other adsorbents reported in the literature.
5. The purity of the NH₃ atmosphere used to prepare La-confined PCMCK-3 should be specified in the Experimental Section.
6. The references are a little outdated. More recent references on template-based fabrication of mesoporous carbon materials, such as Chemical Society Reviews 49, 4681-4736 (2020); Microporous Mesoporous Mater., 307, 110510 (2020); Chemical Engineering Journal 396, 125154 (2020); Journal of Materials Science, 56, 3312-3324 (2021); Electrochimica Acta, 332, 135399 (2020) are recommended to be cited in the Introduction or other appropriate parts.

Reviewer: 2

Comments to the Author(s)

In this paper, the phosphate adsorbents with confined La₂O₃ inside mesoporous carbon were prepared by the solid state grinding method using pristine mesoporous carbon material CMK-3 (PCMCK-3) and oxidized CMK-3 (OCMK-3) as the matrix (denoted as La₂O₃@PCMCK-3 and La₂O₃@OCMK-3), and their adsorption performances for phosphate were investigated. The La-confined adsorbents prepared by the solid state grinding method were well characterized. I think this paper can be considered for publication after major revision. The detail comments are listed as follows:

- (1) It is suggested to assign and mark specific peaks on XRD curves.
- (2) How does the pH affect the adsorption capacity of La₂O₃@ocmk-3 adsorbent? Effect of pH should be discussed, and correlated with Zeta potential, and the pH of solution after adsorption.
- (3) The selectivity of La₂O₃@PCMCK-3 is better, why? Is it related to the structure? Please give a detail explanation.
- (4) The R² values obtained by fitting with Freundlich model are all less than 0.99, especially for La₂O₃ (14.7)@PCMCK-3, which cannot strongly indicate that the adsorption behavior conforms to Freundlich model.
- (5) From Figure 7, it can be seen that the phosphate adsorption capacity of the adsorbent after normalization of La₂O₃ has great difference, which is basically the same as the adsorption capacity described in the paper.
- (6) From Fig. S3., when the loading amount of La₂O₃ in PCMCK-3 is 14.7%, it still has an upward trend compared with the previous point. It is suggested to do two points to ensure that 14.7% is the best loading amounts.
- (7) Please improve language greatly.
- (8) SEM, TEM and corresponding Elemental mapping are necessary to confirm the morphology and the distribution of La₂O₃ on the matrix. Author must provide these data.
- (9) The XRD and XPS analysis of adsorbent after phosphate adsorption should be made to confirm the change of valence of metal element, and also the formation of new crystal phases.

Author's Response to Decision Letter for (RSOS-210428.R0)

See Appendix A.

RSOS-210428.R1 (Revision)

Review form: Reviewer 1

Is the manuscript scientifically sound in its present form?

Yes

Are the interpretations and conclusions justified by the results?

Yes

Is the language acceptable?

Yes

Do you have any ethical concerns with this paper?

No

Have you any concerns about statistical analyses in this paper?

No

Recommendation?

Accept as is

Comments to the Author(s)

The authors have addressed my previous comments thoroughly and conducted additional experiments necessary to improve the quality of this paper. therefore, i am happy to accept the manuscript in the present form.

Review form: Reviewer 2

Is the manuscript scientifically sound in its present form?

Yes

Are the interpretations and conclusions justified by the results?

Yes

Is the language acceptable?

Yes

Do you have any ethical concerns with this paper?

No

Have you any concerns about statistical analyses in this paper?

No

Recommendation?

Accept as is

Comments to the Author(s)

This paper has been revised carefully. Now its quality reaches a publication level. I recommend accept this paper for publication in current form.

Decision letter (RSOS-210428.R1)

Dear Dr Liu:

Title: Confined La₂O₃ particles in mesoporous carbon material for enhanced phosphate adsorption

Manuscript ID: RSOS-210428.R1

It is a pleasure to accept your manuscript in its current form for publication in Royal Society Open Science. The chemistry content of Royal Society Open Science is published in collaboration with the Royal Society of Chemistry.

On behalf of the Subject Editor Professor Anthony Stace and the Associate Editor Professor Chaohua Cui.

RSC Associate Editor:
Comments to the Author:
(There are no comments.)

RSC Subject Editor:
Comments to the Author:
(There are no comments.)

Reviewer(s)' Comments to Author:

Reviewer: 1

Comments to the Author(s)

The authors have addressed my previous comments thoroughly and conducted additional experiments necessary to improve the quality of this paper. therefore, i am happy to accept the manuscript in the present form.

Reviewer: 2

Comments to the Author(s)

This paper has been revised carefully. Now its quality reaches a publication level. I recommend accept this paper for publication in current form.

Appendix A

Dear Editor,

Thank you for your valuable efforts in handling our manuscript. We are also very grateful to the reviewers for their insightful and constructive comments, which significantly improved the quality of our manuscript. The revised manuscript (RSOS-210428), entitled “Confined La_2O_3 particles in mesoporous carbon material for enhanced phosphate adsorption”, has been submitted for review processing. We believe that this revision has addressed the concerns and questions in the reviewers’ comments. Please consult our response to comments on continuous pages regarding the revisions made, which are also highlighted in the revised manuscript for your reference. We hope that our additional data, revisions and the explanations provided are to your satisfaction.

Sincerely yours,

Tao Liu

Reviewer: 1

Comments to the Author(s)

This work reports the encapsulation of lanthanum oxide particles inside pristine and oxidized mesoporous carbon particles by solid state grinding method. The resulting $\text{La}_2\text{O}_3@\text{PCMK-3}$ and $\text{La}_2\text{O}_3@\text{OCMK-3}$ composites have been characterized by different techniques, such as XRD, TEM, BET, FTIR, XRF, Beta-potential, etc. These composites have been tested as adsorbents for phosphate. They found that $\text{La}_2\text{O}_3@\text{OCMK-3}$ samples exhibited the best performance for phosphate adsorption with high adsorption capacity and good regeneration ability. Overall, the work is interesting and explained well. It can be accepted after the following minor revisions:

1. The FT-IR spectra of $\text{La}_2\text{O}_3(14.7)@\text{PCMK-3}$ before and after adsorption should be provided and analyzed.

Response: Thanks for the comment. The FT-IR spectra of $\text{La}_2\text{O}_3(14.7)@\text{PCMK-3}$ before and after adsorption is displayed in Fig. 1. After phosphate adsorption, a strong absorption peak was clearly identified at about 1054 cm^{-1} , which could be assigned to the stretching vibration of P-O of PO_4^{3-} group [1,2]. In addition, The peaks at 613 cm^{-1} and 541 cm^{-1} should be attributed to the bend vibration of O-P-O [3], confirming that phosphate was adsorbed by $\text{La}_2\text{O}_3(14.7)@\text{PCMK-3}$.

Fig. 1. The FT-IR spectra of $\text{La}_2\text{O}_3(14.7)\text{@PCMCK-3}$ before and after adsorption

2. How about the morphology of $\text{La}_2\text{O}_3\text{@PMCK-3}$ after phosphate adsorption? Is the mesoporous structure still maintained?

Response: Thanks for the comment. The morphology of $\text{La}_2\text{O}_3\text{@PMCK-3}$ after phosphate adsorption is presented in Fig. 2, which clearly shows that the morphology of $\text{La}_2\text{O}_3\text{@PMCK-3}$ was well maintained after phosphate adsorption. In addition, compared with the $\text{La}_2\text{O}_3\text{@PMCK-3}$ before phosphate adsorption, the TEM image of $\text{La}_2\text{O}_3\text{@PMCK-3}$ from Fig. 3 shows that the ordered mesoporous structure of PCMCK-3 is still retained after phosphate adsorption.

Fig. 2. The SEM image of $\text{La}_2\text{O}_3\text{@PMCK-3}$ before (a) and after (b) phosphate adsorption

Fig. 3. The TEM image of La₂O₃(14.7)@PCMCK-3 after phosphate adsorption

3. The IR spectra of all La₂O₃@PMCK-3 and La₂O₃@OMCK-3 should be presented rather than general IR spectra of PMCK-3 and OMCK-3, since other characterization data present all the samples.

Response: Thanks for the comment. According to the reviewer's suggestion, the IR spectra of all La₂O₃@PMCK-3 and La₂O₃@OMCK-3 samples were conducted, and the result is shown in Fig. 4. The spectra of all PMCK-3 and OMCK-3 samples showed the bands at 1565 cm⁻¹ and 1130 cm⁻¹, which were assigned to C=C stretching vibrations and skeletal C-C tangential motions, respectively. Compared with PCMCK-3, a weak band at 1715 cm⁻¹ could be observed in the FT-IR spectra of OCMCK-3 samples, which is assigned to C=O stretching vibrations from oxygen containing groups on carbon surface, such as carboxylic acids. A distinct band was observed at 577 cm⁻¹, indicating the presence of La-OH bond vibration [1]. The corresponding revision was also made in the revised manuscript.

Fig. 4. IR spectra of La₂O₃@PMCK-3 and La₂O₃@OMCK-3 samples.

4. Phosphate adsorption performance of the optimum catalyst, La₂O₃(14.7)@PCMCK-3, should be compared with other adsorbents reported in the literature.

Response: Thanks for the comment. According to the reviewer's suggestion, we have summarized some La-based adsorbents reported recently and the result is listed in Table 1. Compared with other La-based adsorbents reported in Table 1, La₂O₃(14.7)@PCMCK-3 and La₂O₃(15.7)@OCMK-3 showed considerable phosphate adsorption capacity. Furthermore, corresponding discussion was also added in the

revised manuscript.

Table 1. Phosphate adsorption capacity of $\text{La}_2\text{O}_3(14.7)\text{@PCMK-3}$,
 $\text{La}_2\text{O}_3(15.7)\text{@OCMK-3}$ and other adsorbents.

Adsorbent	Adsorption capacity (mg/g)	pH	Temperature (°C)	Reference
AL/ Fe_3O_4 / $\text{La}(\text{OH})_3$	60.36	6	25	[4]
AM3	7.08	/	25	[5]
SA-La-5	30.82	6	25	[6]
La_2O_3 /fly ash	24.9	7	25	[7]
$\text{Fe}_3\text{O}_4\text{@MgAl-LDH@La}(\text{OH})_3$	66.5	7	25	[8]
La-modified sludge-based biochar	93.91	5.4	25	[9]
$\text{La}(\text{OH})_3/\text{Fe}_3\text{O}_4$ /bentonite	48.4	7	25	[10]
La_2O_3 /lignocellulosic biochar	36.06	7	25	[11]
$\text{La}(\text{III})/\text{CuFe}_2\text{O}_4$	32.59	7	25	[12]
ACF-LaFe	29.44	6	25	[13]
$\text{La}(\text{OH})_3$ /magnetite	52.7	7	25	[14]
$\text{La}_2\text{O}_3(14.7)\text{@PCMK-3}$	43.67	7	25	This work
$\text{La}_2\text{O}_3(15.7)\text{@OCMK-3}$	47.62	7	25	This work

5. The purity of the NH_3 atmosphere used to prepare La-confined PCMK-3 should be specified in the Experimental Section.

Response: Thanks for the comment. According to the reviewer' comment, we have

revised the experimental expression and the purity in the manuscript.

“Afterwards, the mixture was exposed to NH₃ atmosphere to form hydroxides in a desiccator with 200 mL of concentrated aqueous ammonia (28%) for 48 h.”

6. The references are a little outdated. More recent references on template-based fabrication of mesoporous carbon materials, such as Chemical Society Reviews 49, 4681-4736 (2020); Microporous Mesoporous Mater., 307, 110510 (2020); Chemical Engineering Journal 396, 125154 (2020); Journal of Materials Science, 56, 3312–3324 (2021); Electrochimica Acta, 332, 135399 (2020) are recommended to be cited in the Introduction or other appropriate parts.

Response: Thanks for the comment. According to the reviewer's suggestion, we have added the recent references above in the revised manuscript.

Reviewer: 2

Comments to the Author(s)

In this paper, the phosphate adsorbents with confined La_2O_3 inside mesoporous carbon were prepared by the solid state grinding method using pristine mesoporous carbon material CMK-3 (PCMK-3) and oxidized CMK-3 (OCMK-3) as the matrix (denoted as La_2O_3 @PCMK-3 and La_2O_3 @OCMK-3), and their adsorption performances for phosphate were investigated. The La-confined adsorbents prepared by the solid state grinding method were well characterized. I think this paper can be considered for publication after major revision. The detail comments are listed as follows:

(1) It is suggested to assign and mark specific peaks on XRD curves.

Response: Thanks for the comment. According to the reviewer' comment, we have assigned and marked the specific peaks on XRD curves (see Fig. 5).

Fig. 5. Small-angle XRD patterns of the samples.

(2) How does the pH affect the adsorption capacity of La₂O₃@OCMK-3 adsorbent?

Effect of pH should be discussed, and correlated with Zeta potential, and the pH of solution after adsorption.

Response: Thanks for the comment. According to the reviewer's comment, the pH influence on phosphate adsorption on La₂O₃(15.7)@OCMK-3 adsorbent was conducted and the result is displayed in Fig. 6. The pH values were tested after adsorption equilibrium. It could be obviously observed that the phosphate adsorption

capacity of $\text{La}_2\text{O}_3(15.7)\text{@OCMK-3}$ decreased with the increasing of final pH of solution. According to the reviewer's comment, we have added more discussion about pH influence on phosphate adsorption in the revised manuscript.

“With pH increasing from 3 to 11, the surface charge of adsorbents became more negative and then the repulsive force towards phosphate dominated. Notably, with pH increasing from 3 to 11, the zeta potentials of $\text{La}_2\text{O}_3(14.7)\text{@PCMK-3}$ and $\text{La}_2\text{O}_3(15.7)\text{@OCMK-3}$ decreased from 5.63 mV to -24.2 mV and -1.9 mV to -22.17 mV, respectively. Therefore, at high pH the negatively charged surface was expected to inhibit the adsorption of phosphate on active sites, which led to continuously decreased phosphate adsorption with the final pH of solution.”

Fig. 6 The pH influence on phosphate adsorption on $\text{La}_2\text{O}_3(15.7)\text{@OCMK-3}$

(3) The selectivity of $\text{La}_2\text{O}_3\text{@PCMK-3}$ is better, why? Is it related to the structure? Please give a detail explanation.

Response: Thanks for the comment. The mechanism of phosphate adsorption on La-based adsorbents could mainly be ascribed to ligand exchange by forming

inner-sphere complexes. The schematic description of the mechanism for phosphate adsorption on $\text{La}_2\text{O}_3@\text{PCMK-3}$ is illustrated in Fig. 7. For phosphate adsorption, the adsorbed species phosphate anions are directly bound to the lanthanum hydroxides, and inner-sphere complexes are formed. Consequently, the hydroxyls from La-OH groups of adsorbents could be exchanged by phosphate. However, for Cl^- , NO_3^- and SO_4^{2-} anions, the combination with the lanthanum hydroxides was mainly dependent on the electrostatic interaction. Compared with the chemical bond of phosphate and lanthanum, the interaction between Cl^- , NO_3^- , SO_4^{2-} and lanthanum hydroxide is relatively weaker. Therefore, even coexisting with high concentration anions, phosphate showed the greater competitiveness for active sites, and $\text{La}_2\text{O}_3@\text{PCMK-3}$ showed highly selective phosphate adsorption performance. In addition, previous studies also reported that different La-based adsorbents showed the similar phosphate adsorption selectivity, such as pure lanthanum hydroxide [15], La-loaded type adsorbent [1] and La-confined type adsorbent [16]. In this work, due to its large surface area and ordered pore structure, mesoporous carbon material CMK-3 was applied as carrier to enhance the dispersion of La_2O_3 particles. Therefore, we believed that $\text{La}_2\text{O}_3@\text{OCMK-3}$ should also exhibit higher adsorption selectivity towards phosphate than coexisting Cl^- , NO_3^- and SO_4^{2-} anions.

Fig. 7. The schematic representation of the mechanism for phosphate adsorption on $\text{La}_2\text{O}_3(14.7)\text{@PCMK-3}$

(4) The R^2 values obtained by fitting with Freundlich model are all less than 0.99, especially for $\text{La}_2\text{O}_3(14.7)\text{@PCMK-3}$, which cannot strongly indicate that the adsorption behavior conforms to Freundlich model.

Response: Thanks for the comment. According to the reviewer's comment (6), for the isotherm experiments of $\text{La}_2\text{O}_3(14.7)\text{@PCMK-3}$ and $\text{La}_2\text{O}_3(15.7)\text{@OCMK-3}$, two more points were conducted to ensure the best phosphate adsorption amounts respectively. The result of re-fitted parameters is listed in Table 2. The correlation coefficient values R^2 were calculated to be 0.83 to 0.98. To relieve reviewer's concern,

we have revised the expression more accurately.

“Table 2 lists the parameters fitted by the Freundlich model, which could well describe phosphate adsorption on the adsorbents.”

Table 2. Isotherm parameters for phosphate adsorption on the adsorbents fitted by Freundlich model.

Samples	$1/n$	K_F (mg P ¹⁻ⁿ /(L ⁿ ·g))	R^2
La ₂ O ₃ (6.8)@PCMk-3	0.25	6.84	0.94
La ₂ O ₃ (12.1)@PCMk-3	0.14	15.32	0.83
La ₂ O ₃ (14.7)@PCMk-3	0.18	21.23	0.89
La ₂ O ₃ (6.9)@OPCMk-3	0.22	9.50	0.88
La ₂ O ₃ (12.1)@OCMk-3	0.12	19.97	0.98
La ₂ O ₃ (15.7)@OCMk-3	0.16	27.25	0.93

(5) From Figure 7, it can be seen that the phosphate adsorption capacity of the adsorbent after normalization of La₂O₃ has great difference, which is basically the same as the adsorption capacity described in the paper.

Response: Thanks for the comment. Compared with the isotherms in Fig. 6 of the original manuscript, the phosphate adsorption capacities of adsorbents after La₂O₃ normalization in Fig.7 of the original manuscript was relatively distributed within a smaller area. For example, at equilibrium phosphate concentration of about 1.5 mg/L, the phosphate adsorption capacities of La₂O₃(6.8)@PCMk-3, La₂O₃(12.1)@PCMk-3 and La₂O₃(14.7)@PCMk-3 were 8.77 mg/g, 19.32 mg/g and 22.53 mg/g, respectively.

After La_2O_3 normalization, the phosphate adsorption capacities of $\text{La}_2\text{O}_3(6.8)\text{@PCMK-3}$, $\text{La}_2\text{O}_3(12.1)\text{@PCMK-3}$ and $\text{La}_2\text{O}_3(14.7)\text{@PCMK-3}$ were calculated to be 128.85 mg/g, 159.67 mg/g and 153.78 mg/g. Therefore, compared with the data distributions of phosphate adsorption capacity in Fig. 6, the adsorption capacity data distributions of $\text{La}_2\text{O}_3\text{@PCMK-3}$ and $\text{La}_2\text{O}_3\text{@OCMK-3}$ adsorbents in Fig. 7 could be considered to be approximately identical. The main reason was that compared with pure La_2O_3 , the La_2O_3 particles were well dispersed and confined in $\text{La}_2\text{O}_3\text{@PCMK-3}$ and $\text{La}_2\text{O}_3\text{@OCMK-3}$ adsorbents, which led to higher La_2O_3 availability and identical phosphate adsorption capacity after La_2O_3 normalization. To relieve reviewer's concern, the discussion in this section of manuscript is revised now to express more accurate. The sentence is revised to "*Despite different La_2O_3 loading amounts, the La_2O_3 normalized phosphate adsorption capacities of $\text{La}_2\text{O}_3\text{@PCMK-3}$ adsorbents showed a comparable trend within tested phosphate concentrations (see Fig. 7). Similarly, the $\text{La}_2\text{O}_3\text{@OCMK-3}$ adsorbent showed the same result.*"

(6) From Fig. S3., when the loading amount of La_2O_3 in PCMK-3 is 14.7%, it still has an upward trend compared with the previous point. It is suggested to do two points to ensure that 14.7% is the best loading amounts.

Response: Appreciate the comment. According to the reviewer's suggestion, the phosphate adsorption isotherms with higher initial phosphate concentrations were conducted to clarify the adsorption capacity, and the result is illustrated in Fig. 8. It showed that the adsorption did increase with the increasing of initial phosphate

concentration as the reviewer suggested. Accordingly, the corresponding revision has also been made in the manuscript.

Fig.8. Isotherm of phosphate adsorption on $\text{La}_2\text{O}_3\text{@PCMk-3}$

(7) Please improve language greatly.

Response: Thanks for the comment. According to the reviewer’s suggestion, we have invited a native English speaker to improve the language and tried our best to meet the requirements of the journal.

(8) SEM, TEM and corresponding Elemental mapping are necessary to confirm the morphology and the distribution of La_2O_3 on the matrix. Author must provide these data.

Response: Thanks for the comment. The SEM images of PCMk-3 and $\text{La}_2\text{O}_3(14.7)\text{@PCMk-3}$ are illustrated in Fig. 9. From Fig. 9, it could be observed that the morphology of PCMk-3 was well maintained after lanthanum oxide loading.

In order to confirm the distribution of La in the adsorbent

$\text{La}_2\text{O}_3(14.7)\text{@PCMK-3}$, TEM-EDX analysis was conducted, and the result is illustrated in Fig. 10. The corresponding EDX mapping showed La was evenly distributed in $\text{La}_2\text{O}_3(14.7)\text{@PCMK-3}$ (Fig. 10d), confirming La species were effectively confined in PCMK-3 mesopores. In addition, P elements are perfectly coordinating to the distribution of La elements in Fig. 10e, indicating that La was the active species responsible for phosphate adsorption.

Fig. 9. The SEM images of PCMK-3 (a) and $\text{La}_2\text{O}_3(14.7)\text{@PCMK-3}$ (b)

Fig. 10. The TEM image of $\text{La}_2\text{O}_3(14.7)\text{@PCMk-3}$ after phosphate adsorption (a).

The EDX mapping selected area (b) and the EDX images of C (c), La (d) and P (e).

(9) The XRD and XPS analysis of adsorbent after phosphate adsorption should be made to confirm the change of valence of metal element, and also the formation of

new crystal phases.

Response: Thanks for the comment. According to the reviewer's comment, the XRD patterns of $\text{La}_2\text{O}_3(14.7)\text{@PCMK-3}$ adsorbent after phosphate adsorption were analyzed and displayed in Fig. 11. Compared with $\text{La}_2\text{O}_3(14.7)\text{@PCMK-3}$ before phosphate adsorption, new diffraction peaks were observed at $2\theta = 19.9^\circ$, 31.0° and 41.8° from the pattern of $\text{La}_2\text{O}_3(14.7)\text{@PCMK-3}$ after phosphate adsorption, which were corresponding to the facet of $\text{LaPO}_4/\text{LaPO}_4 \cdot 0.5\text{H}_2\text{O}$ (JCPDS No. 75-1881/46-1439) [17], indicating that phosphate was adsorbed on $\text{La}_2\text{O}_3(14.7)\text{@PCMK-3}$.

Fig. 11. The XRD patterns of $\text{La}_2\text{O}_3(14.7)\text{@PCMK-3}$ before and after adsorption. ★ represents the characteristic diffraction peaks of $\text{LaPO}_4/\text{LaPO}_4 \cdot 0.5\text{H}_2\text{O}$

To further clarify the composition change of $\text{La}_2\text{O}_3(14.7)\text{@PCMK-3}$ before and after adsorption, the high resolution XPS spectra of $\text{La}_2\text{O}_3(14.7)\text{@PCMK-3}$ were conducted and displayed in Fig. 12. For the high resolution spectra of La 3d before adsorption, the peaks at 835.75 eV and 852.38 eV could be ascribed to La_2O_3 , and the peaks at 838.80 eV and 855.63 eV were corresponding to $\text{La}(\text{OH})_3$ [18]. After

phosphate adsorption, a slight positive binding energy shift (~ 0.625 eV) of La_2O_3 was determined, indicating the formation of LaPO_4 complex [19]. For P 2p scanning spectra in Fig. 12b, the peak at binding energy of 132.8 eV corresponding to P 2p was clearly identified after phosphate adsorption, indicating the formation of LaPO_4 compound [20]. Accordingly, the corresponding revision was also made in the revised manuscript.

Fig. 12. High resolution XPS spectra of La 3d (a) and P 2p (b) of $\text{La}_2\text{O}_3(14.7)@\text{PCMK-3}$ before and after adsorption.

Reference:

1. Liu JY, Zhou Q, Chen, JH, Zhang L, Chang N. 2013 Phosphate adsorption on hydroxyl-iron-lanthanum doped activated carbon fiber. *Chem. Eng. J.* **215–216**, 859–867.
2. Soejoko DS, Tjia MO. 2003 Infrared spectroscopy and X ray diffraction study on the morphological variations of carbonate and phosphate compounds in giant prawn (*Macrobrachium rosenbergii*) skeletons during its moulting period. *J. Mater. Sci.* **38**, 2087–2093.
3. Li L, Jiang W, Pan H, Xu X, Tang Y, Ming J, Xu Z, Tang R. 2007 Improved luminescence of lanthanide (III)-doped nanophosphors by linear aggregation. *J. Phys. Chem. C.* **111**, 4111–4115.
4. Li CS, Li YJ, Li Q, Duan JL, Hou JY, Hou Q, Ai SY, Li HS, Yang YC. 2021 Regenerable magnetic aminated lignin/Fe₃O₄/La(OH)₃ adsorbents for the effective removal of phosphate and glyphosate. *Sci. Total. Environ.* **788**, 147812.
5. Jin HY, Lin L, Meng XY, Wang LL, Huang Z, Liu M, Dong L, Hu Y, Crittenden JC. 2021 A novel lanthanum-modified copper tailings adsorbent for phosphate removal from water. *Chemosphere* **281**, 130779
6. Yang XY, Wei YM, Jiang YH, Wang YY, Chen L, Peng L, Zhang S, Yan Y, Yan YS. 2021 High efficiency phosphate removal was achieved by lanthanum modified mesoporous silica aerogels with cellulose-guided templates. *Ind. Eng. Chem. Res.* **60**, 5352–5363.
7. Asaoka S, Kawakami K, Saito H, Ichinari T, Nohara H, Oikawa T. 2021 Adsorption

of phosphate onto lanthanum-doped coal fly ash-Blast furnace cement composite. *J. Hazard. Mater.* **406**, 124780.

8. Lin Z, Chen J. 2021 Magnetic $\text{Fe}_3\text{O}_4@\text{MgAl-LDH}@\text{La}(\text{OH})_3$ composites with a hierarchical core-shell structure for phosphate removal from wastewater and inhibition of labile sedimentary phosphorus release. *Chemosphere*, **264**, 128551.

9. Li J, Li B, Huang H, Zhao N, Zhang M, Cao L. 2020. Investigation into lanthanum-coated biochar obtained from urban dewatered sewage sludge for enhanced phosphate adsorption. *Sci. Total. Environ.* **714**, 136839.

10. Zhong Z, Lu X, Yan R, Lin S, Wu X, Huang M, Liu Z, Zhang F, Zhang B, Zhu H, Guo X. 2020 Phosphate sequestration by magnetic La-impregnated bentonite granules: a combined experimental and DFT study. *Sci. Total Environ.* **738**, 139636.

11. Xu Q, Chen Z, Wu Z, Xu F, Yang D, He Q, Li G, Chen Y. 2019 Novel lanthanum doped biochars derived from lignocellulosic wastes for efficient phosphate removal and regeneration. *Bioresource Technol.* **289**, 121600.

12. Gu W, Li X, Xing M, Fang W, Wu D. 2018 Removal of phosphate from water by amine-functionalized copper ferrite chelated with La(III). *Sci. Total. Environ.* **619-620**, 42–48.

13. Liu X, Zong E, Hu W, Song P, Wang J, Liu Q, Ma Z, Fu S. 2018 Lignin-derived porous carbon loaded with $\text{La}(\text{OH})_3$ nanorods for highly efficient removal of phosphate. *ACS Sustain. Chem. Eng.* **7(1)**, 758–768.

14. Fang L, Liu R, Li J, Xu C, Huang LZ, Wang D. 2018 Magnetite/lanthanum hydroxide for phosphate sequestration and recovery from lake and the attenuation

effects of sediment particles. *Water Res.* **130**, 243–254.

15. Xie J, Wang Z, Lu SY, Wu DY, Zhang ZJ, Kong HN. 2014 Removal and recovery of phosphate from water by lanthanum hydroxide materials. *Chem. Eng. J.* **254**, 163–170.

16. Zhang YY, Pan, BC, Shan, C, Gao X. 2016 Enhanced phosphate removal by nanosized hydrated La(III) oxide confined in cross-linked polystyrene networks. *Environ. Sci. Technol.* **50**, 1447–1454.

17. Zhang Y, Wang M, Gao X, Qian J, Pan BC. 2021 Structural evolution of lanthanum hydroxides during long-term phosphate mitigation: effect of nanoconfinement. *Environ. Sci. Technol.* **55**, 665-676.

18. Li JPH, Zhou XH, Pang YQ, Zhu L, Vovk EI, Cong LN, van Bavel AP, Li SG, Yang Y. 2019 Understanding of binding energy calibration in XPS of lanthanum oxide by *in situ* treatment. *Phys. Chem. Chem. Phys.* **21**, 22351.

19. Wu B, Fang L, Fortner JD, Guan X, Lo IMC, 2017. Highly efficient and selective phosphate removal from wastewater by magnetically recoverable La(OH)₃/Fe₃O₄ nanocomposites. *Water Res.* **126**, 179–188.

20. Liu R, Li J, Xu C, Huang LZ, Wang D. 2018. Magnetite/Lanthanum hydroxide for phosphate sequestration and recovery from lake and the attenuation effects of sediment particles. *Water Res.* **130**, 243–254.